# A Data-Driven Comprehensive Battery SOH Evaluation and Prediction Method Based on Improved CRITIC-GRA and Att-BiGRU

**Peng Liu [1], Cheng Liu [1], Zhenpo Wang [1], Qiushi Wang [1,*] , Jinlei Han [2] and Yapeng Zhou [3]**

[1] National Engineering Research Center of Electric Vehicles, School of Mechanical Engineering, Beijing Institute of Technology, Beijing 100081, China; bitliupeng@bit.edu.cn (P.L.); liucheng981127@163.com (C.L.)
[2] Fawer Smarter Energy Technology Company Limited, Jilin 130062, China; hanjinlei@fawer.com.cn
[3] China Merchants Testing Vehicle Technology Research Institute Co., Ltd., Chongqing 401329, China; peng66886688@163.com
* Correspondence: wangqiushi@bit.edu.cn

**Abstract:** The state-of-health (SOH) of lithium-ion batteries has a significant impact on the safety and reliability of electric vehicles. However, existing research on battery SOH estimation mainly relies on laboratory battery data and does not take into account the multi-faceted nature of battery aging, which limits the comprehensive and effective evaluation and prediction of battery health in real-world applications. To address these limitations, this study utilizes real electric vehicle operational data to propose a comprehensive battery health evaluation indicator and a deep learning predictive model. In this study, the battery capacity, ohmic resistance, and maximum output power were initially extracted as individual health indicators from actual vehicle operation data. Subsequently, a methodology that combines the improved criteria importance through inter-criteria correlation (CRITIC) weighting method with the grey relational analysis (GRA) method is employed to construct the comprehensive battery health evaluation indicator. Finally, a prediction model based on the attention mechanism and the bidirectional gated recurrent unit (Att-BiGRU) is proposed to forecast the comprehensive evaluation indicator. Experimental results using real-world vehicle data demonstrate that the proposed comprehensive health indicator can provide a thorough representation of the battery health state. Furthermore, the Att-BiGRU prediction model outperforms traditional machine learning models in terms of prediction accuracy.

**Keywords:** lithium-ion batteries; health estimation; comprehensive evaluation; deep learning

## 1. Introduction

In response to global climate change, the energy crisis, and the pursuit of low-carbon goals, the development of electric vehicles has become an inevitable trend in the global automotive industry [1–4]. With the advent of vehicle electrification, the driving range and safety of electric vehicles have become major concerns. Lithium-ion batteries (LIBs), responsible for energy storage and supply, are core components of electric vehicles, and their performance and health state have significant impacts on the reliability and safety of electric vehicles [5–7]. Although LIBs have long lifespans, they experience varying degrees of aging and degradation with increasing usage time and cycles [8–10]. Therefore, accurately assessing and predicting the state-of-health (SOH) of batteries has become a research hotspot.

Currently, the SOH of LIBs is mainly represented by single features, which can be classified into capacity-based methods, internal resistance-based methods, and power-based methods [11]. The capacity-based method represents SOH by the ratio of the current capacity to the initial capacity of the battery. The internal resistance-based method reflects

battery health by measuring the growth rate of the battery's ohmic resistance. The power-based method characterizes battery health by assessing the power state during charge and discharge at specific state-of-charge (SOC) levels [12,13].

Methods for estimating battery SOH can be categorized into experimental methods, model-based methods, and data-driven methods [14,15]. Experimental methods involve testing batteries under controlled laboratory conditions, but they have limitations such as long testing cycles and differences between laboratory conditions and real-world operating conditions, making them unsuitable for online estimation of SOH in real-world vehicle applications [16]. Model-based methods rely on mathematical models, electrochemical models, or equivalent circuit models to simulate the dynamic response characteristics of batteries and identify battery parameters for health estimation [17–19]. However, model-based methods struggle to accurately simulate the internal working characteristics of batteries, which limits their accuracy and practicality for real-world battery systems. In contrast, data-driven methods extract battery health feature parameters directly from large amounts of data without the need for a detailed understanding of battery chemistry. These methods then establish a mapping relationship between the extracted features and SOH through model training to achieve health state estimation [20,21]. For instance, Lin et al. constructed an SOH estimator using an explainable boosting machine based on the Oxford dataset by extracting features such as internal resistance and thermal–electric coupling [22]. Xiong et al. proposed a feature extraction method that combines multiple algorithms to extract the most relevant features from different voltage ranges using laboratory battery aging data, and a machine learning algorithm was used to estimate SOH [23]. Tian et al. proposed an SOH prediction model based on an end-to-end deep convolutional neural network, which used short-time charging features extracted from small windows and directly mapped battery capacity [24]. These data-driven methods have shown promising prediction performance for battery SOH.

However, several challenges still exist for battery SOH evaluation and prediction for real-world electric vehicles. First, battery ageing is a comprehensive process with manifestations in many aspects, existing SOH representation methods for batteries that consider only a single factor fail to comprehensively evaluate the various aspects of battery performance, which do not meet the practical requirements of electric vehicle applications. Furthermore, there are no effective weighing methods proposed for battery multi-parameter health indicator evaluation, which is imperative for scoring the overall battery performance and remaining value. Lastly, most of these methods are established under well-controlled laboratory conditions that are quite different from the time-varying EV operating conditions, which raises concerns about their performance for real-world battery SOH estimation.

To address the above issues, this paper proposes a comprehensive evaluation and prediction method for battery health based on real-world vehicle data. The main contributions of the paper are as follows:

(1)  Based on the characteristics of real-world EV data, basic health indicators including capacity, ohmic resistance, and maximum output power are extracted using specific methods suitable for EV application scenarios.

(2)  An improved criteria importance through the inter-criteria correlation (CRITIC) weighting method is introduced in order to obtain objective weights for three typical battery health indicators. These weights are then combined with the grey relational analysis (GRA) method to construct a comprehensive evaluation indicator for battery health.

(3)  Leveraging the advantages of bidirectional gated recurrent unit (BiGRU) and attention mechanism, an Att-BiGRU deep learning model is developed to predict the comprehensive health state of batteries.

The overall research framework of this paper is shown in Figure 1. The remainder of this paper is organized as follows: Section 2 describes the data used in this study and the methods for extracting health indicators. Section 3 presents the calculation method for the comprehensive battery health indicator and introduces the developed Att-BiGRU

prediction model. Section 4 validates the effectiveness and accuracy of the proposed methods. Finally, Section 5 concludes the paper.

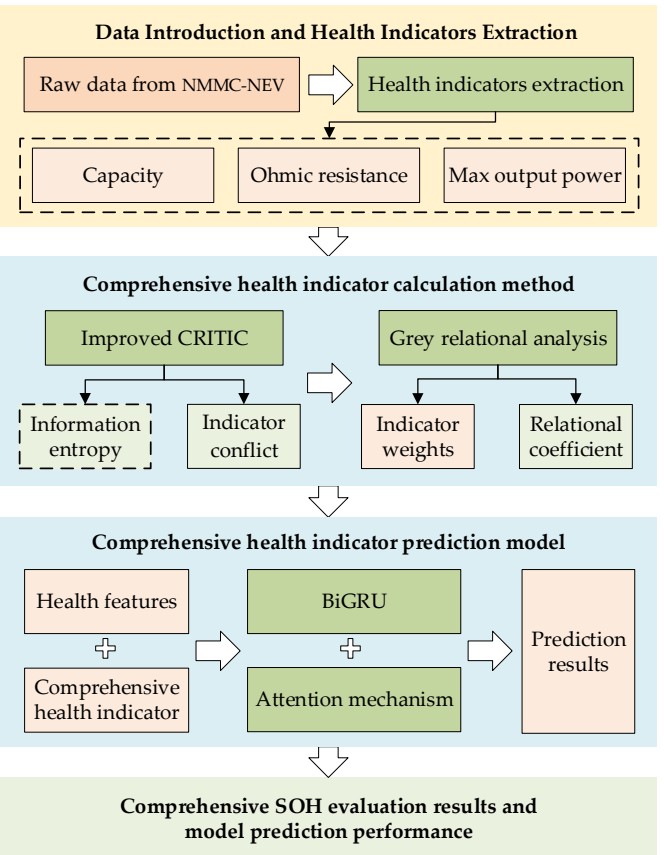

**Figure 1.** The overall research framework of the study.

## 2. Data Introduction and Health Indicators Extraction

### 2.1. Data Introduction

The data used in this study are collected from the open lab of the National Monitoring and Management Center for New Energy Vehicles (NMMC-NEV), which records the actual operating data of over 10 million new energy vehicles. The platform provides real-time data including time, vehicle speed, cumulative mileage, total voltage, total current, SOC, temperature, and other variables. The sampling frequency of these data is 0.1 Hz, which can accurately reflect the vehicle and battery states.

Due to their regular driving patterns and long travel distances, electric buses are suitable for analyzing battery health. Therefore, this study selects multiple electric buses of a certain model from the NMMC-NEV, with each bus having an accumulated mileage exceeding 150,000 km. The specifications of the selected vehicles are presented in Table 1. The selected vehicles were put into operation in Guangdong and Liaoning provinces in May 2017. To ensure the amount of data, the actual operating data of the selected vehicles from May 2017 to May 2021 are used to conduct a comprehensive health evaluation and prediction study in this article. Data preprocessing methods such as data smoothing, and data interpolating are implemented to clean the dataset, which is a crucial preliminary for ensuring the quality of the dataset.

**Table 1.** The specifications of the selected vehicles.

| Parameter | Value |
|---|---|
| Cathode material | LiFePO$_4$ |
| Battery capacity | 240 Ah |
| Driving range | 300 km |
| Motor power | 100 kW |
| Curb weight | 8500 kg |

*2.2. Extraction of Health Representation Indicators*

Through the analysis in Section 1, we know that capacity, internal resistance, and power are commonly used battery health evaluation indicators in previous studies. Therefore, in this section, we select these three single indicators as the basic indicators for comprehensive battery health evaluation. The specific extraction methods of different indicators are as follows.

2.2.1. Capacity

In the actual operation of electric vehicles, it is rare to have complete charge and discharge cycles, making it difficult to calculate the actual capacity of the battery. To ensure an adequate sample size and the effectiveness of capacity analysis, this study characterizes the battery's capacity degradation by calculating the regional charging capacity within a certain SOC interval, as described in [25]. As shown in Figure 2, the fitted curve indicates that the regional capacity of the battery gradually decreases with increasing cumulative mileage, indicating a decline in the actual capacity of the battery. Additionally, capacity is an important performance parameter of batteries that directly affects the driving range of electric vehicles. Therefore, capacity can be used as an indicator to evaluate the health state of batteries.

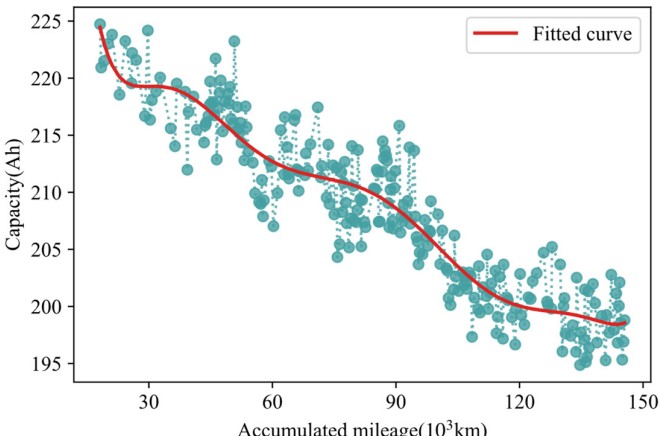

**Figure 2.** The battery capacity degrades with increasing accumulated mileage.

2.2.2. Ohmic Resistance

Due to the unavailability of direct access to the battery's ohmic resistance from the large data platform, this study employs a first-order equivalent circuit model to identify the ohmic resistance. The identification results are shown in Figure 3. It can be observed that the ohmic resistance of the battery exhibits periodic variations with increasing cumulative mileage, showing an overall increasing trend. Additionally, the ohmic resistance is significantly influenced by temperature, with higher resistance values observed in the low-temperature range compared to the high-temperature range. After fitting the ohmic resistance at 27 °C and 37 °C, it is found that the resistance growth rate is higher in the low-temperature range. Therefore, the ohmic resistance demonstrates periodic changes

with the aging of batteries and can be used as an indicator to evaluate the health state of batteries.

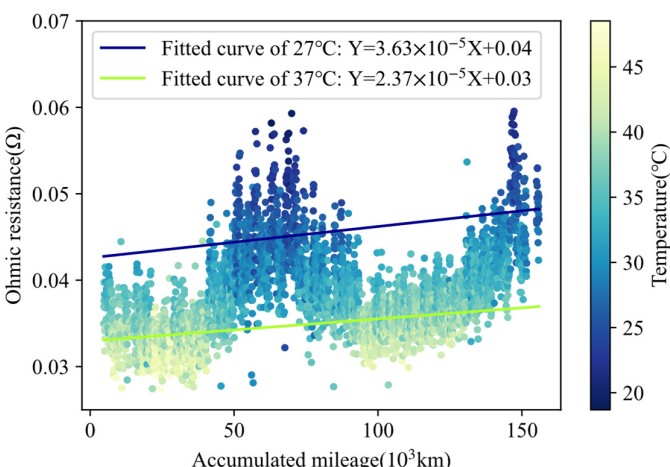

**Figure 3.** The ohmic resistance of the battery exhibits changes with increasing accumulated mileage.

### 2.2.3. Maximum Output Power

Since it is not feasible to directly conduct maximum output power tests on operating vehicles, this study calculates the maximum output power within each driving segment based on the power definition. Then, the average of the maximum output power of each driving segment within each 1000-km interval is calculated as the maximum output power of the battery over a long-time duration. Figure 4 illustrates that the maximum output power of the selected electric buses decreases most significantly at high temperatures with increasing cumulative mileage, exhibiting an overall declining trend. Additionally, the power output of the battery has an impact on the vehicle's driving performance. Therefore, the maximum power output can be used as an evaluation indicator for the health state of the battery.

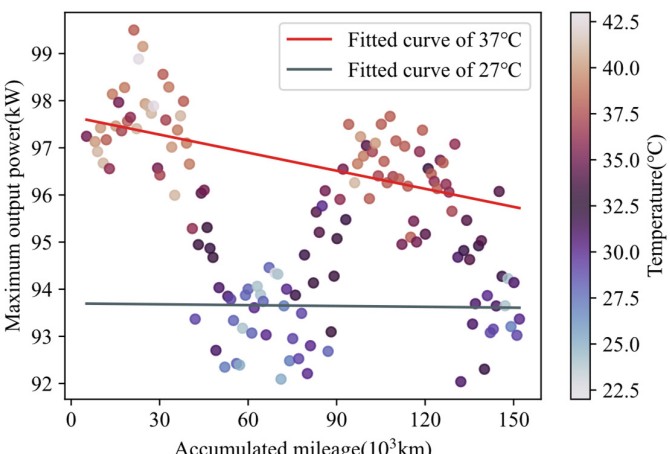

**Figure 4.** The maximum output power of the battery decreases with increasing accumulated mileage.

## 3. Methods

### 3.1. Comprehensive Battery Health Indicator Based on Improved CRITIC and GRA

#### 3.1.1. Improved CRITIC Weighting Method

The CRITIC method is an objective weighting approach that simultaneously considers the impact of data variability within indicators and the interrelationships among indicators. Specifically, it employs two key concepts to determine weights: the contrast intensity and the conflicting character of the evaluation criteria [26]. First, if an indicator exhibits

significant differences in values across various samples, it possesses a higher contrast intensity, resulting in a larger weight. Second, if an indicator shows stronger correlations with other indicators, it has less conflict with them, resulting in a smaller weight. Generally, standard deviation is used to measure the contrast intensity, while correlation coefficients are employed to assess the conflict among indicators.

Due to the limitations of standard deviation in fully capturing the information within a single indicator, as it mainly measures the dispersion and fluctuation of data, this study proposes a modified method using information entropy instead of standard deviation to represent the comparative strength of data within each indicator. Information entropy can provide a more comprehensive quantification of the information content within each indicator, allowing for a more accurate evaluation of the comparative strength of indicators [27].

The improved CRITIC weighting method consists of the following steps:

1. Data normalization

In this paper, different sequences of battery health indicators during total life cycles are normalized in two cases. For indicators such as capacity and maximum output power that decrease gradually with battery aging, the following formula is used for normalization:

$$x_{ij}^* = \frac{x_{ij} - \min_i\{x_{ij}\}}{\max_i\{x_{ij}\} - \min_i\{x_{ij}\}}, \tag{1}$$

where $i$ is the number of life cycles and $j$ is the number of health indicators.

For indicators such as ohmic internal resistance, which increases gradually with battery aging, the following formula is used:

$$x_{ij}^* = \frac{\max_i\{x_{ij}\} - x_{ij}}{\max_i\{x_{ij}\} - \min_i\{x_{ij}\}}. \tag{2}$$

2. Calculate the comparative strength of indicators based on information entropy

First, the entropy value of each metric is calculated as shown in the following equation:

$$h_j = -\frac{1}{\ln m}\sum_{i=1}^{m} p_{ij}\ln(p_{ij}), \tag{3}$$

where

$$p_{ij} = \frac{x_{ij}^*}{\sum\limits_{i=1}^{m} x_{ij}^*}. \tag{4}$$

The more information the data contains, the lower the entropy. Therefore, in the improved CRITIC weighting method, the comparative strength can be represented as

$$H_j = 1 - h_j. \tag{5}$$

3. Calculate the conflict between indicators

A stronger correlation between a single indicator and the other indicators indicates a higher degree of duplicated information and a lower level of conflict. Therefore, the quantification formula for conflict is as follows:

$$R_j = \sum_{k=1}^{n}\left(1 - r_{jk}\right), \tag{6}$$

where $r_{jk}$ is the correlation between the $j$th indicator and the other indicators.

4. Calculate the weights for each indicator

Firstly, we calculate the amount of comprehensive information for each indicator using

$$C_j = H_j \times R_j. \tag{7}$$

Then, the weights of the indicators are calculated in the following equation:

$$w_j = \frac{C_j}{\sum\limits_{i=1}^{n} C_j}. \tag{8}$$

### 3.1.2. GRA Comprehensive Evaluation Method

GRA is a comprehensive evaluation method used for multi-criteria decision analysis. It leverages the principle of correlation, measuring the degree of similarity between a reference sequence and a comparison sequence [28]. By quantifying the grey relational coefficient, GRA provides a numerical indicator of the relationship strength between variables.

The GRA method mainly consists of the following steps:

1.  Construct the evaluation matrix

In this study, three health evaluation indicators were selected. Here, we have constructed a scoring matrix for these different evaluation indicators at varying accumulated mileage points. This matrix is subjected to the same data normalization method as used in the improved CRITIC. The resulting normalized scoring matrix is presented below:

$$\mathbf{Z} = \begin{bmatrix} z_{11} & z_{12} & z_{13} \\ z_{21} & z_{22} & z_{23} \\ \vdots & \vdots & \vdots \\ z_{m1} & z_{m2} & z_{m3} \end{bmatrix}, \tag{9}$$

where each column represents a health evaluation indicator, and each row represents the values of all indicators at different cumulative mileage.

2.  Determine the reference sequence

The reference sequence is derived from the normalized matrix, which is generally the optimal value of each column. Following the application of indicator normalization, this paper selects the maximum values from each column as the reference sequence.

The formula for the positive ideal solution is expressed as follows:

$$\mathbf{Z}_0 = (z_1, z_2, z_3), \tag{10}$$

where

$$z_j = \max\{z_{1j}, z_{2j}, \ldots, z_{mj}\}. \tag{11}$$

3.  Calculate the grey relational coefficient

The gray correlation coefficient is used to determine the closeness of each comparison sequence to the reference sequence. The specific formula is defined as

$$\gamma\left(z_{1j}, z_{ij}\right) = \frac{\min\limits_{i}\min\limits_{j}\Delta_{ij} + \rho\max\limits_{i}\max\limits_{j}\Delta_{ij}}{\Delta_{ij} + \rho\max\limits_{i}\max\limits_{j}\Delta_{ij}}, \tag{12}$$

where $\rho$ is the distinguishing coefficient, and $\Delta_{ij} = \left|z_{1j} - z_{ij}\right|$ is the difference matrix.

4.  Calculate the grey relational grade

The gray relational grade (GRG) is used to measure the similarity between different comparison sequences and the reference sequence. Different health indicators have different contributions to the comprehensive evaluation results. Therefore, this paper integrates the

weight of the health indicator into the CRG; that is, it uses the weighted CRG to determine the comprehensive battery SOH. The calculation method of the weighted CRG is as follows:

$$r_i = \sum_{j=1}^{n} w_j \gamma \left( z_{1j}, z_{ij} \right), \tag{13}$$

where $w_j$ represents the weight of each health indicator, which can be obtained using the improved CRITIC weighting method proposed in this study.

### 3.1.3. The Improved CRITIC-GRA Method

To achieve a more objective and comprehensive battery SOH evaluation, this section combines the improved CRITIC weighting method and the GRA method to propose a comprehensive battery health evaluation indicator that considers multiple single battery health indicators.

First, an evaluation indicator matrix is constructed based on the extracted single battery health indicator data; then, the improved CRITIC weighting method is used to calculate the objective weights of different battery health indicators and the indicator weight matrix is constructed; then, the GRA is used to calculate the gray correlation coefficients of different health indicators under different degradation conditions and construct the gray correlation coefficient matrix; finally, the objective weights of different indicators are integrated into the quantification of gray correlation degrees; that is, the gray correlation coefficient matrix is multiplied by the objective weight matrix of single health indicators to obtain the final comprehensive evaluation indicator. The calculation formula of the battery health comprehensive evaluation indicator proposed in this section is as follows:

$$\mathbf{Co\_SOH} = \mathbf{\Gamma} \times \mathbf{W^T} = \begin{bmatrix} \gamma_{11} & \gamma_{12} & \gamma_{13} \\ \gamma_{21} & \gamma_{22} & \gamma_{23} \\ \vdots & \vdots & \vdots \\ \gamma_{m1} & \gamma_{m2} & \gamma_{m3} \end{bmatrix} \times \begin{bmatrix} w_1 \\ w_2 \\ w_3 \end{bmatrix}, \tag{14}$$

where $\mathbf{\Gamma}$ is the gray correlation coefficient matrix and $\mathbf{W^T}$ is the objective weight matrix.

### 3.2. Battery Comprehensive Health Prediction Model Based on Att-BiGRU

### 3.2.1. Feature Extraction for Model Input

The input features of the model play a crucial role in influencing the prediction performance. Initially, during the process of health indicator extraction, it was observed that there exists a strong correlation between cumulative mileage and battery health indicators. Furthermore, temperature also directly affects certain indicators. Therefore, this study directly extracted two health features, cumulative mileage and temperature, from the raw data variables. Moreover, research suggests that SOC and charging current are vital factors impacting battery health. To comprehensively exploit real-world vehicle data and ensure accurate predictions by the model, this study further computes and extracts 8 health features from the charging segments, focusing on SOC, voltage, and current.

Consequently, a total of 10 health features were obtained, as presented in Table 2, thereby accounting for the multifaceted aspects of battery health evaluation.

**Table 2.** Extracted health features.

| Number | Feature | Number | Feature |
|--------|---------|--------|---------|
| F1 | Accumulated mileage | F6 | Charging start voltage |
| F2 | Temperature | F7 | Charging end voltage |
| F3 | Charging start SOC | F8 | Charging voltage difference |
| F4 | Charging end SOC | F9 | Maximum charging current |
| F5 | Charging SOC difference | F10 | Average charging current |

### 3.2.2. Att-BiGRU

The gated recurrent unit (GRU) is an improved version of recurrent neural networks (RNNs). GRU introduces two gating mechanisms, namely, the reset gate and the update gate, to regulate the flow of information within the hidden state. This design mitigates the challenges of gradient vanishing or exploding encountered by l RNNs during long-term memory and backpropagation. The fundamental computational unit of GRU is outlined in Figure 5.

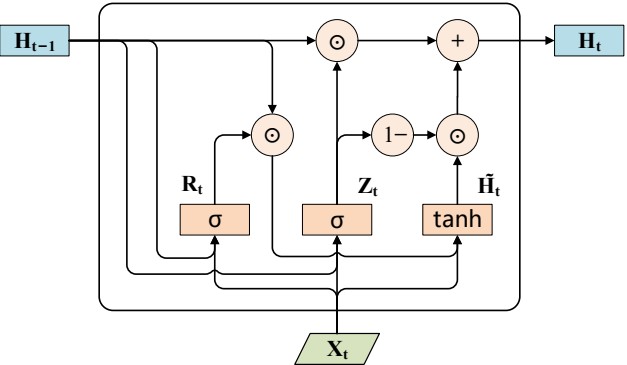

**Figure 5.** The fundamental computational unit of GRU.

The BiGRU model is a bidirectional recurrent neural network consisting of both forward and backward GRU layers. The forward GRU layer processes input sequences in chronological order, while the backward GRU layer processes input sequences in reverse chronological order. At time step "t", the hidden state of the BiGRU is obtained by concatenating the forward and backward hidden states. This approach enables the model to simultaneously incorporate past and future information, facilitating a more effective modeling of inter-sequence dependencies.

The attention mechanism is a widely employed technique in tasks such as natural language processing and machine translation, aiming to model the varying degrees of focus on different segments within sequences. Traditional sequence models employ fixed-weight parameters to process each element within a sequence. However, this fixed-weight approach may fall short of effectively capturing vital contextual information for longer sequences or intricate semantic relationships. In contrast, the attention mechanism employs dynamic weights that allow the model to adaptively allocate more attention to crucial segments, depending on the task's requirements.

While BiGRU offers some improvement in addressing the issue of long-distance dependencies in RNNs, its information retention capability still remains limited. This study introduces a fusion model of the BiGRU and the attention mechanism, aiming to address these limitations. The proposed approach involves employing the attention mechanism to dynamically weight the hidden states of BiGRU at each step, thereby directing attention towards hidden states that hold greater significance for predictive outcomes. The Att-BiGRU model inherits the advantages of BiGRU while simultaneously mitigating the loss of node information during the prediction process for long sequences.

The constructed Att-BiGRU prediction model in this study is illustrated in Figure 6.

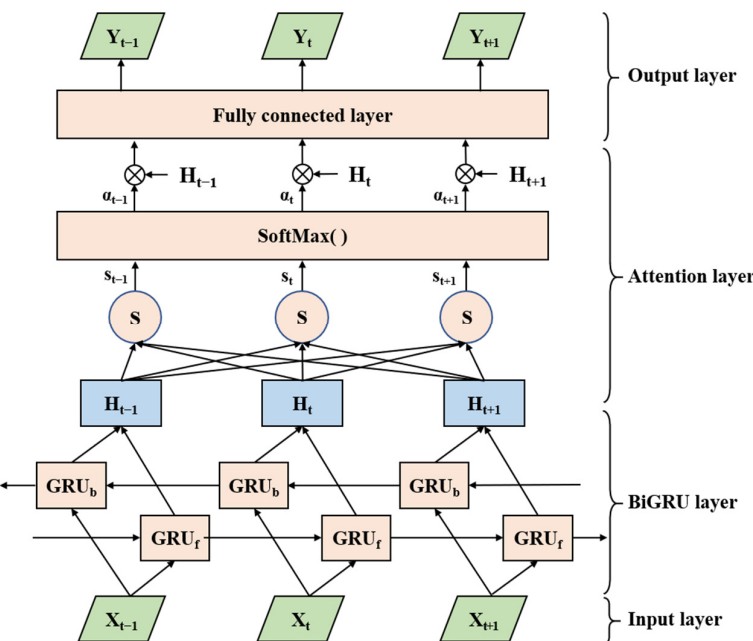

**Figure 6.** The structure of Att-BiGRU.

## 4. Results and Discussion

All the results in this section are obtained based on a personal computer equipped with an AMD Ryzen 7 4800U CPU, an AMD Radeon Graphics, and 16 GB RAM (Santa Clara, CA, USA). The deep learning models run on the PyTorch framework based on Python 3.8.

### 4.1. Results of Comprehensive Battery Health Evaluation

In this section, we select four electric buses of the same model in actual operation to conduct verification and discussion of the proposed method. Vehicle A and Vehicle B operate in Shenyang with an average temperature of 10.1 °C, while Vehicle C and Vehicle D operate in Guangzhou with an average temperature of 21.9 °C.

First, we used the method in Section 2.2 to extract health indicator data for the four vehicles. Then, based on the extracted battery health indicator data, the improved CRITIC weighting method was employed to ascertain the weights of each indicator, as presented in Table 3. The weights shed light on the relative importance of different health indicators for each bus. For Vehicle A and Vehicle B, the internal resistance carries the highest weight at 0.41 and 0.48, which could potentially be attributed to the lower average temperature in the city where they operate. This lower temperature might lead to a faster increase in internal resistance due to battery aging. For Vehicle C and Vehicle D, the capacity indicator holds the highest weight at 0.53 and 0.54. However, it does not exhibit overwhelming dominance when compared to the other two indicators. In conclusion, these weight distributions underscore the limitation of relying solely on a single health indicator to depict the battery's health state, as it might present an overly narrow perspective.

**Table 3.** The weights of each indicator.

| Vehicle | Capacity | Resistance | Power |
|---------|----------|------------|-------|
| A | 0.38 | 0.41 | 0.21 |
| B | 0.34 | 0.48 | 0.18 |
| C | 0.53 | 0.21 | 0.26 |
| D | 0.54 | 0.24 | 0.22 |

Subsequently, the GRA method was employed to calculate the comprehensive battery health indicators for both buses. In Figure 7, the curve labeled "SOH" represents the battery SOH based on capacity and the "Co_SOH" represents the comprehensive health indicator, while the "D-value" represents the difference between them. It can be observed that for all four vehicles, the values of comprehensive health indicators obtained in this study differ from the capacity-based SOH by approximately ±4%. For Vehicle A, prior to reaching 90,000 km, the Co_SOH is slightly higher than the SOH. Afterwards, the Co_SOH experiences an accelerated decline, becoming lower than the SOH, and the difference between them gradually increases. Similarly, for Vehicle B, up to 70,000 km, the Co_SOH and SOH are relatively close, with Co_SOH being slightly lower than SOH. Afterwards, Co_SOH experiences an accelerated decline, and the difference between them gradually increases. The changing trends in Co_SOH for Vehicle A and Vehicle B are indeed primarily related to the higher weight assigned to ohmic resistance compared to capacity in the evaluation process. In contrast, Vehicles C and D exhibit similar patterns, where Co_SOH is slightly higher than SOH, and the difference accumulates as the mileage increases. This is primarily due to the lower weight assigned to ohmic resistance and maximum output power compared to capacity in the evaluation process. Conclusively, the proposed comprehensive health indicator in this study provides a more comprehensive representation of the battery health state, effectively capturing various degradation aspects.

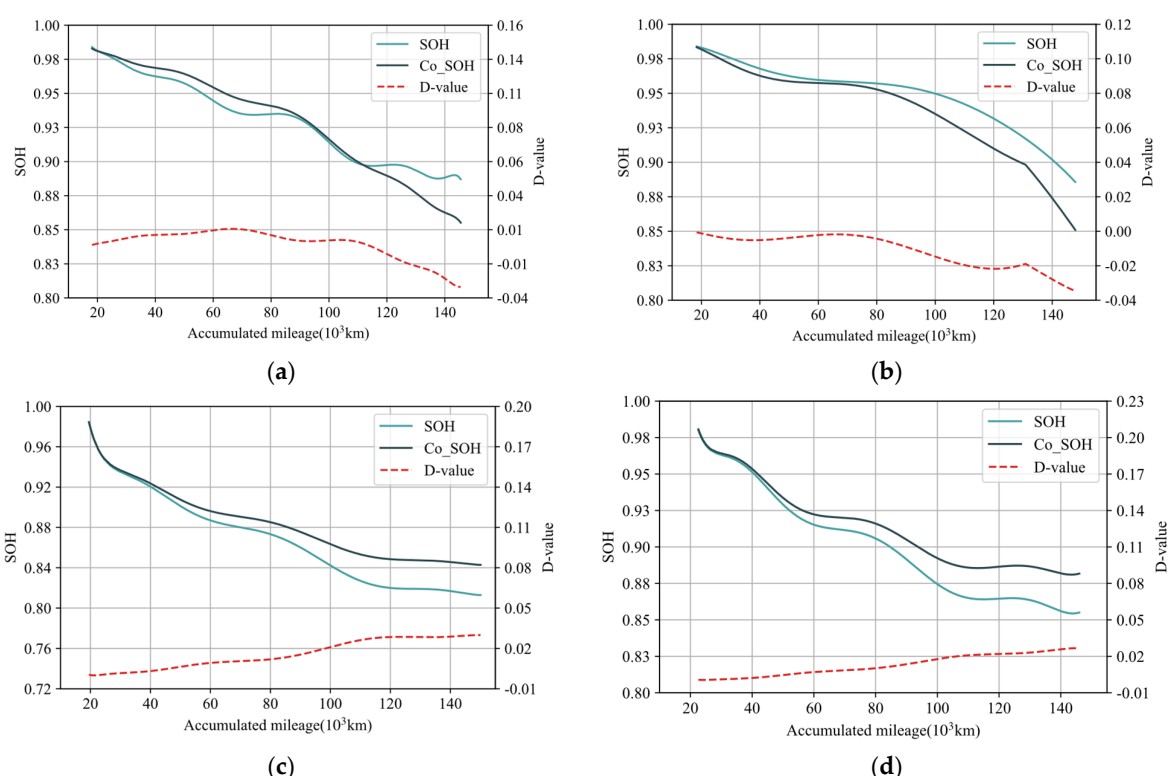

**Figure 7.** Comprehensive indicator of battery health: (**a**) Vehicle A; (**b**) Vehicle B; (**c**) Vehicle C; (**d**) Vehicle D.

### 4.2. Prediction of Comprehensive Health Indicator

In this study, the prediction of the comprehensive battery health indicator is treated as a time series forecasting task. The prediction utilizes data considered as time series data, where each row represents a time frame. Therefore, in this section, according to the order of accumulated mileage from small to large, the 10 health features initially selected in Section 3.2 are used as input features, and the comprehensive health indicator obtained in Section 3.1 is used as a label to construct the data set used in the Att-BiGRU model. The first 70% of the dataset is the training set, and the last 30% is the test set.

In order to ensure prediction accuracy and improve prediction speed, this section conducts a correlation analysis on the initially selected 10 health features in this study. Features with an absolute value of correlation coefficient greater than 0.5 with the comprehensive health indicator of the battery are selected as input features for the prediction model. Taking Vehicle A as an example, the correlation of each feature with the comprehensive health indicator is shown in Figure 8.

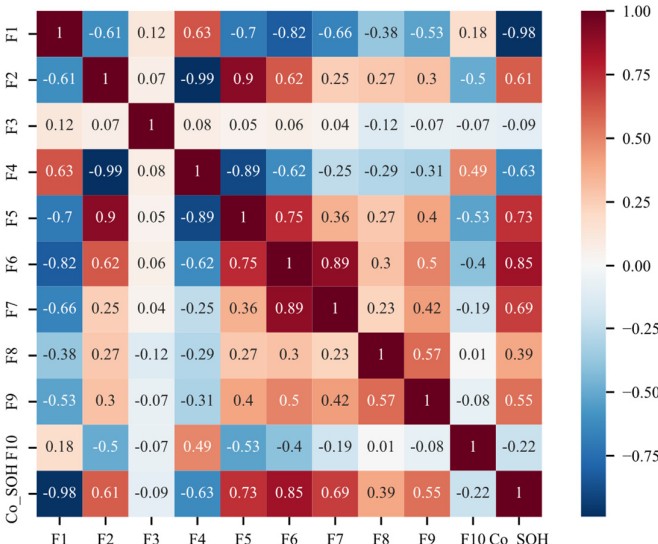

**Figure 8.** Correlation of each feature with the comprehensive health indicator.

Here, the time step of the Att-BiGRU model is set to 1; that is, the data of the previous segment is used to predict the comprehensive health of the current segment. To assess the prediction performance of the Att-BiGRU model, a comparative analysis is conducted against XGBoost, GRU, and BiGRU models. The predicted values are presented in Figure 9, and the prediction performance is summarized in Table 4.

From Figure 9, it is evident that for all vehicles, the Att-BiGRU model exhibits a commendable ability to fit the original comprehensive health indicator values. In contrast to the classic ensemble learning approach, XGBoost, the predictive curves of Att-BiGRU, GRU, and BiGRU models display reduced volatility, closely aligning with the true values.

As presented in Table 4, across the prediction of different vehicle data, the Att-BiGRU model outperforms other models in terms of root mean square error (RMSE), mean absolute error (MAE) and $R^2$. With the largest RMSE and MAE, XGBoost consistently demonstrates the least accuracy among the models. Deep learning methods such as GRU, BiGRU, and especially Att-BiGRU exhibit a marked improvement in predictive accuracy compared to XGBoost. While BiGRU offers a slight advantage over GRU, the enhancement is moderate. Incorporating the attention mechanism into BiGRU significantly enhances predictive accuracy and reduces prediction errors. For Vehicle A, Att-BiGRU yields an RMSE of 0.056, representing a 32% reduction compared to BiGRU. Furthermore, Att-BiGRU's MAE of 0.048 is 29% lower than BiGRU. For Vehicle B, the RMSE of Att-BiGRU is 0.071, which is 28% lower than BiGRU. In addition, Att-BiGRU's MAE of 0.063 is 26% lower than BiGRU. For Vehicle C, the RMSE of Att-BiGRU is 0.070, which is 27% lower than BiGRU. In addition, Att-BiGRU's MAE of 0.045 is 35% lower than BiGRU. Similarly, for Vehicle D, Att-BiGRU achieves an RMSE of 0.032, 29% lower than BiGRU and an MAE of 0.025, indicating a 32% improvement over BiGRU. In addition, Att-BiGRU achieves the largest $R^2$ values among all models in all four vehicles, reaching more than 0.956, which means that the prediction results fit the real curve well. Especially compared with XGBoost, the $R^2$ value has been improved by 6% on average. Hence, the employed Att-BiGRU model for comprehensive battery health indicator prediction proves to be highly effective.

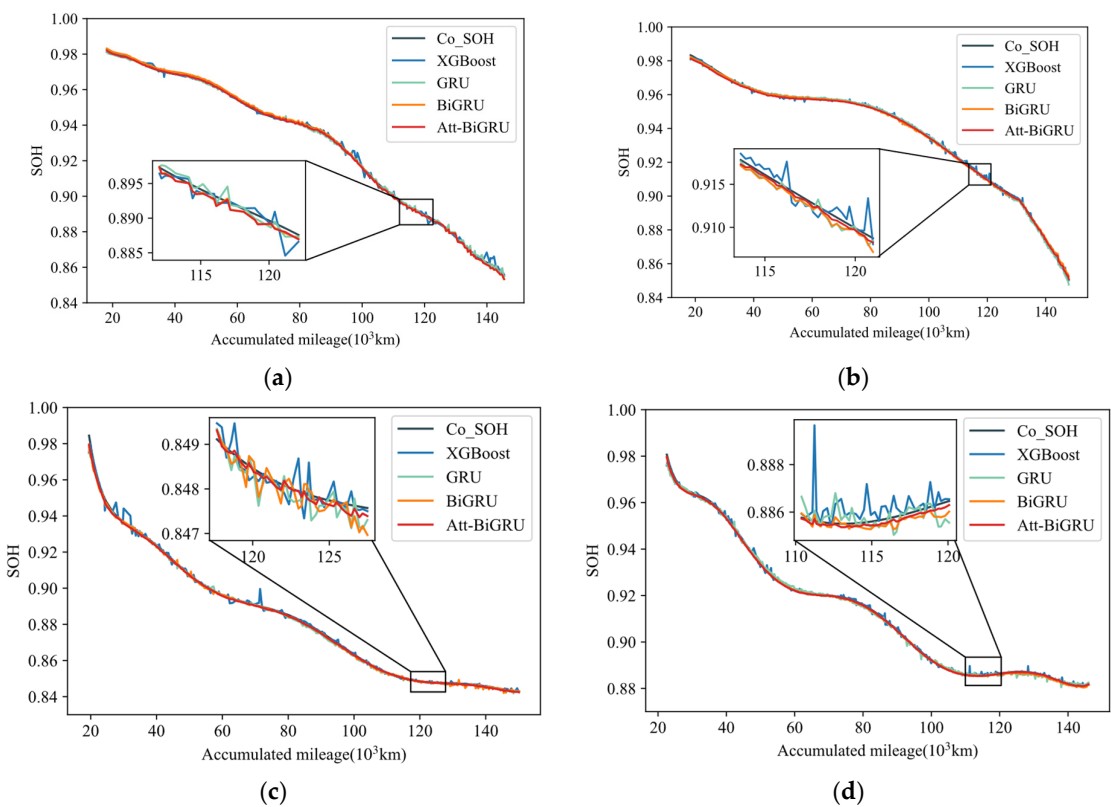

**Figure 9.** Prediction results of Att-BiGRU model: (**a**) Vehicle A; (**b**) Vehicle B; (**c**) Vehicle C; (**d**) Vehicle D.

**Table 4.** Prediction performance of the comprehensive health indicator across different models.

| Model | A | | | B | | | C | | | D | | |
|---|---|---|---|---|---|---|---|---|---|---|---|---|
| | RMSE | MAE | $R^2$ | RMSE | MAE | $R^2$ | RMSE | MAE | $R^2$ | RMSE | MAE | $R^2$ |
| XGB | 0.132 | 0.097 | 0.924 | 0.205 | 0.150 | 0.910 | 0.214 | 0.128 | 0.908 | 0.116 | 0.081 | 0.927 |
| GRU | 0.094 | 0.072 | 0.942 | 0.107 | 0.088 | 0.928 | 0.106 | 0.081 | 0.933 | 0.087 | 0.067 | 0.945 |
| BiGRU | 0.082 | 0.068 | 0.951 | 0.098 | 0.085 | 0.937 | 0.096 | 0.069 | 0.949 | 0.045 | 0.037 | 0.972 |
| Att-BiGRU | 0.056 | 0.048 | 0.973 | 0.071 | 0.063 | 0.956 | 0.070 | 0.045 | 0.971 | 0.032 | 0.025 | 0.985 |

## 5. Conclusions

In order to address the limitation of the single indicator in battery health evaluation, this paper proposes a comprehensive evaluation and prediction method considering multiple health indicators based on real-world vehicle data. Initially, common battery health evaluation indicators are extracted, and their evolution characteristics with battery aging are analyzed. Objective weights for each evaluation indicator are determined using the improved CRITIC weighting method, followed by the derivation of a comprehensive health indicator through the integration of GRA algorithm. Subsequently, the Att-BiGRU model is employed to predict the proposed comprehensive health indicator. Comparative analyses are conducted against other predictive models. The introduced comprehensive health indicator offers a more holistic evaluation of battery health state and demonstrates good interpretability. The predictive model utilizing Att-BiGRU for the comprehensive health indicator attains a low RMSE of only 0.032 and a MAE of 0.025, both outperforming algorithms such as XGBoost, GRU, and BiGRU in terms of prediction accuracy. In conclusion, this study makes valuable contributions by offering a multi-faceted battery health evaluation framework and an accurate predictive model, ultimately contributing to a more comprehensive understanding of battery health state evaluation.

In future research, we will extract more health state evaluation indicators based on actual vehicle operating data to further improve the comprehensiveness of the battery

health comprehensive evaluation indicator proposed in this article. In addition, in the future, we will further improve the practicality of the method proposed in this article and try to carry out real vehicle deployment.

**Author Contributions:** Conceptualization, P.L. and C.L.; methodology, C.L.; software, C.L.; validation, C.L.; formal analysis, P.L. and Q.W.; investigation, C.L.; resources, P.L., Z.W., J.H. and Y.Z.; data curation, C.L.; writing—original draft preparation, C.L.; writing—review and editing, Q.W.; visualization, C.L.; supervision, P.L.; project administration, P.L. and Q.W.; funding acquisition, P.L., Z.W., J.H. and Y.Z. All authors have read and agreed to the published version of the manuscript.

**Funding:** This research was funded by the Jilin Scientific and Technological Development Program (No. 20220301019GX) and the Chongqing Doctoral Through-Train Research Project (No. sl202100000050).

**Institutional Review Board Statement:** Not applicable.

**Informed Consent Statement:** Not applicable.

**Data Availability Statement:** Data sharing is not applicable to this article.

**Conflicts of Interest:** The authors declare no conflict of interest.

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
