# Peer review of "A Data-Driven Comprehensive Battery SOH Evaluation and Prediction Method Based on Improved CRITIC-GRA and Att-BiGRU"

_sustainability, doi:10.3390/su152015084_

Round 1

Reviewer 1 Report

The manuscript is well written and adequate results are discussed. The present form of the manuscript can be accepted for publication.

Minor English correction is required.

Author Response

Thanks a lot for your kindly reminder.  We have improved the English expression of the manuscript.

Reviewer 2 Report

In this paper, a method is proposed for evaluating and predicting multiple health indicators based on real-world vehicle data. The method takes into consideration various factors and aims to provide a comprehensive evaluation. To begin with, the paper focuses on extracting common battery health evaluation indicators. Furthermore, the evolution characteristics of these indicators with battery aging are thoroughly analyzed.

- In page 12, "The predicted values are presented in Figure 10 ...", It seems wrong figure number. please correct it. 

The quality of English language is good.  

Author Response

Thanks a lot for your kindly reminder.  We have corrected the wrong figure number in page 12.

Reviewer 3 Report

Can be improved for the betterment of the manuscript and for the research community
